# Calpain-3 Is Not a Sodium Dependent Protease and Simply Requires Calcium for Activation

**DOI:** 10.3390/ijms24119405

**Published:** 2023-05-28

**Authors:** Stefan G. Wette, Graham D. Lamb, Robyn M. Murphy

**Affiliations:** 1Department of Biochemistry and Chemistry, La Trobe Institute for Molecular Science, School of Agriculture, Biomedicine and Environment, La Trobe University, Melbourne, VIC 3086, Australia; s.wette@latrobe.edu.au; 2Department of Microbiology, Anatomy, Physiology and Pharmacology, School of Agriculture, Biomedicine and Environment, La Trobe University, Melbourne, VIC 3086, Australia; g.lamb@latrobe.edu.au

**Keywords:** ryanodine receptor, junctophilin, titin fragments, skeletal muscle, proteolytic activity

## Abstract

Calpain-3 (CAPN3) is a muscle-specific member of the calpain family of Ca^2+^-dependent proteases. It has been reported that CAPN3 can also be autolytically activated by Na^+^ ions in the absence of Ca^2+^, although this was only shown under non-physiological ionic conditions. Here we confirm that CAPN3 does undergo autolysis in the presence of high [Na^+^], but this only occurred if all K^+^ normally present in a muscle cell was absent, and it did not occur even in 36 mM Na^+^, higher than what would ever be reached in exercising muscle if normal [K^+^] was present. CAPN3 in human muscle homogenates was autolytically activated by Ca^2+^, with ~50% CAPN3 autolysing in 60 min in the presence of 2 µM Ca^2+^. In comparison, autolytic activation of CAPN1 required about 5-fold higher [Ca^2+^] in the same conditions and tissue. After it was autolysed, CAPN3 unbound from its tight binding on titin and became diffusible, but only if the autolysis led to complete removal of the IS1 inhibitory peptide within CAPN3, reducing the C-terminal fragment to 55 kDa. Contrary to a previous report, activation of CAPN3, either by raised [Ca^2+^] or Na^+^ treatment, did not cause proteolysis of the skeletal muscle Ca^2+^ release channel-ryanodine receptor, RyR1, in physiological ionic conditions. Treatment of human muscle homogenates with high [Ca^2+^] caused autolytic activation of CAPN1, accompanied by proteolysis of some titin and complete proteolysis of junctophilin (JP1, full length ~95 kDa), generating an equimolar amount of a diffusible ~75 kDa N-terminal JP1 fragment, but without any proteolysis of RyR1.

## 1. Introduction

Calpains are calcium-dependent proteases that are involved in a wide range of cellular processes. There are at least 15 calpain isoforms identified in mammals. Of particular interest in skeletal muscle are the ubiquitously expressed calpain-1 and calpain-2, also known as mu-calpain and m-calpain that consensus nomenclature now refers to as CAPN1 and CAPN2, respectively, as well as a muscle-specific calpain, calpain-3 (CAPN3). Investigations into the differences between the calpains have focused on the calcium concentration required for their activation and, in the case of CAPN1 and CAPN3, their autolysis. Proteolytic activity of native 94 kDa CAPN3 is normally prevented by the presence of an internal 48-residue insertion sequence, IS1, which stops assembly of the critical catalytic residues [1]. The binding of calcium ions to two non-EF-hand Ca^2+^ binding sites enables intramolecular autolysis, first at the N-terminal end of IS1 and subsequently its C-terminal end, leaving a C-terminal segment of CAPN3 of 60, 58 and then 55 kDa [2,3]. The N- and C-terminal sections of the autolysed CAPN3 molecule self-associate to form the active protease [4]. Critically, CAPN3 is activated at low physiological calcium concentrations ([Ca^2+^]), with recombinant CAPN3 seen to rapidly (~5 min) autolyse at ~500 nM Ca^2+^ [5] and native CAPN3 in rat skeletal muscle fibres found to undergo ~20% autolysis upon exposure to an intracellular [Ca^2+^] of 200 nM for 60 min, with this being reliant on the co-presence of physiological ATP levels [6]. Importantly, mutations in the gene encoding CAPN3 can give rise to limb-girdle muscular dystrophy type 2A, a progressive muscle-wasting disease characterized by weakness and atrophy of the proximal limb muscles.

Although the physiological role of CAPN3 remains undefined, independent studies have shown that virtually all CAPN3 in muscle is tightly bound to structural proteins [6], predominantly on titin at the N2A line but also on the M-line [6,7]. The association with titin is believed to stabilize the CAPN3 [8], although no ‘spontaneous’ autolysis of CAPN3 was observed over >60 min, even if it was freely diffusible in solution, provided that the bathing solution mimicked normal intracellular conditions and the free [Ca^2+^] was tightly maintained at or below the normal resting level (~50 nM) [6].

Intriguingly, it has been reported that CAPN3 can also be activated by raised intracellular levels of sodium (Na^+^) and thus it has been referred to as being an intracellular Na^+^-dependent protease [9]. However, in that work, the [Na^+^] used was much higher than would ever occur intracellularly in skeletal muscle fibres, and importantly, the ionic constitution of the solutions used were not physiological; in particular, they completely lacked all K^+^. The physiological concentration range of intracellular [Na^+^] in skeletal muscle is ~10 mM (in a resting muscle fibre) to an upper concentration of ~25–30 mM (in a highly stimulated muscle fibre) [10,11]. Ionic strength is predominantly contributed to by the intracellular [K^+^], which is present at ~130–150 mM.

Given that calpains are traditionally regarded as Ca^2+^-dependent proteases, we explored this suggested Na^+^-dependency using physiologically relevant [Na^+^] for the autolysis, and hence activation, of endogenously expressed CAPN3 in rat and human skeletal muscle under heavily buffered calcium conditions. We also determined what effects there might be when other non-physiological solutions are used for examining CAPN3 autolysis and whether the CAPN3 redistributed to the cytosol when autolysed. Finally, we also examined whether autolytic activation of CAPN3 by either Na^+^ or Ca^2+^ treatments led to proteolysis of the ryanodine receptor (RyR1) in skeletal muscle (from rat and human), the vital Ca^2+^ release channel in the sarcoplasmic reticulum, which has been reported to be a proteolytic target of CAPN3 [12].

## 2. Results

### 2.1. Effect of Na^+^ on CAPN3 Autolysis

Rat skeletal muscle tissue was exposed for various times (0–90 min) to a physiologically based intracellular solution (physiological in terms of intracellular ionic strength, osmolality, pH, [K^+^], [Na^+^], ATP and free Mg^2+^), with the [Ca^2+^] heavily buffered with 50 mM EGTA to maintain very low [Ca^2+^] (<10 nM). Under these conditions, there was little apparent autolysis of CAPN3 even over 90 min (Figure 1), in agreement with our previous observations [6,13]. Importantly, there was no increase in CAPN3 autolysis regardless of whether the [Na^+^] in the solution was close to the level in a rested muscle fibre (K^+^-based solution with 10 mM Na^+^) or was slightly above the highest level the [Na^+^] might reach with intense muscle activity (K^+^-based solution with 36 mM Na^+^) (Figure 1B,C). In marked contrast, if the [Na^+^] was increased to such an extent that it completely replaced all the K^+^ normally present in the bathing solution (162 mM Na^+^, 0 mM K^+^) (i.e., Na^+^-based with no K^+^ solution), CAPN3 autolysed in a time-dependent manner, with ~60% of the total CAPN3 found in its 55 kDa autolysed form after 90 min (*p* < 0.05, two-way; Figure 1B,C).

We then utilised the mechanically skinned fibre technique to examine whether the autolysis of CAPN3 occurring in the Na^+^-based solution with no K^+^ resulted in any redistribution of the CAPN3 between the fibre and the cytosol (Figure 2). We found that exposure to 162 mM Na^+^ for 60 min resulted in the redistribution of autolysed CAPN3 to the cytosol (i.e., wash/W in Figure 2A). Given our finding that CAPN3 autolysed in a time-dependent manner when exposed to the Na^+^-based solution with no K^+^, we hypothesised that this could be a consequence of the solution having no K^+^ present rather than being due to the presence of the high [Na^+^], particularly given that there was no autolysis in the presence of 36 mM Na^+^ when a high [K^+^] was still present. To further investigate this hypothesis, we examined the effect of another condition with zero [K^+^] (primarily Tris-Cl, with ~16 mM Na^+^) on CAPN3 autolysis. Similar to results with 162 mM Na^+^, CAPN3 autolysis also occurred the Tris-Cl solution with no K^+^ (Figure 2B, right).

### 2.2. Ca^2+^-Dependence of CAPN3 Autolysis, and Redistribution of Autolysed CAPN3 to Cytosol

We have previously shown with rat skinned muscle fibres that under normal intracellular physiological conditions (i.e., in K^+^-based solution with 36 mM Na^+^), CAPN3 autolysis occurs if the free [Ca^2+^] is raised slightly above its normal resting level of ~50 nM to ~200 nM, with ~20% autolysis seen after 60 min [6]. Here, we investigated the Ca^2+^-dependence of CAPN3 autolysis in human skeletal muscle samples in the same physiological conditions and found similar characteristics as seen for rat muscle, with appreciable CAPN3 autolysis evident after a 60 min exposure to a [Ca^2+^] of 2–500 μM (Figure 3).

The experiments also examined the Ca^2+^-dependence of autolysis of CAPN1 in the same tissue samples and conditions (Figure 3A, lowest panel). This revealed that CAPN1 autolysis also showed a tight Ca^2+^-dependence but required ~5 times higher [Ca^2+^] under the same conditions to elicit a similar degree of autolysis (Figure 3B). Analysis of the curve-fitting models revealed that there was a 99.5% probability that the data support different curves rather than one curve for both data sets. Next, we used the 500 µM Ca^2+^ solution producing maximal CAPN3 autolysis to examine whether the localisation of CAPN3 was altered upon autolysis with such Ca^2+^ treatment (Figure 3C) similar to that seen with treatment with 162 mM Na^+^. Compared to Control samples (where [Ca^2+^] was kept <10 nM), after 60 min of Ca^2+^ treatment virtually all the CAPN3 had autolysed and most had redistributed to the cytosolic/Cyt fraction (*n* = 4, *p* < 0.05, two-way ANOVA; Figure 3D,E).

The broad ionic dependence of CAPN1 and CAPN3 autolysis, which is similar in rat and human skeletal muscle is presented in Table 1.

### 2.3. Further Examination of Effect of Na^+^ Treatment on CAPN3 Localisation

Given that Na^+^ treatment of rat skinned fibres (Figure 2B) and high [Ca^2+^] treatment of human muscle samples (Figure 3C–E) both resulted in autolysis of CAPN3 and its redistribution to the cytosol, we carried out more detailed fractionation procedures on Na^+^-treated human muscle samples to determine whether the cytosolic relocation of autolysed CAPN3 was a robust phenomenon. In these experiments, a portion of the K^+^-treated and Na^+^-treated whole muscle samples (Wh) were each separated into three fractions: a cytosolic fraction (Cyt), a membrane-associated fraction (Membrane or Me) and the remaining cytoskeletal (and nuclear) fraction (Csk) (see Section 4 and Figure 4A). The effectiveness of the fractionation procedure was confirmed by a) almost all GAPDH being in the Cyt fraction, b) almost all SERCA2a being in the Me fraction and c) all the myosin being in the Csk fraction (Figure 4A) (see also [14]). Consistent with the results presented above (Figure 1 and Figure 2), the Na^+^ treatment caused a large increase in CAPN3 autolysis, as seen in the Wh samples (*n* = 4, *p* = 0.02, paired *t*-test; Figure 4A,B). The fractionation procedure showed that over 60% of the CAPN3 that had autolysed with the Na^+^ treatment redistributed to the Cyt and Me fractions (*p* < 0.05, two-way ANOVA; Figure 4A,D), whereas very little of either the autolysed or the non-autolysed CAPN3 present in the control samples, treated under physiological K^+^ conditions, translocated to the Cyt or Me fractions. Interestingly, CAPN1 autolysis and localisation were unaffected by exposure to the Na^+^-based solution with no K^+^ (*p* > 0.05, two-way ANOVA; Figure 4E,F).

### 2.4. RyR1 Protein Is Not Proteolyzed upon CAPN3 Autolysis with Treatment with High [Na^+^] or 500 µM Ca^2+^

CAPN3 has been observed to proteolyze a variety of structural proteins in various in-vitro conditions [16] but the physiological or pathological relevance of such findings are presently unclear. It has also been reported that CAPN3 can proteolyze the ryanodine receptor Ca^2+^ release channel (RyR1) in skeletal muscle fibres [12], but those experiments were not carried out with physiological ionic conditions. Here, we investigated whether RyR1 was proteolyzed upon autolysis of CAPN3 by either Na^+^ treatment or raised [Ca^2+^]. Na^+^ treatment of rat skinned fibres resulted in the expected autolysis of CAPN3 and its unbinding and movement to the cytoplasm, but this was not accompanied by any detectable proteolysis of RyR1 (Figure 5A, left panel), with there being no increase in any RyR1 proteolytic products in the range from 180 to 550 kDa and no apparent loss of the RyR1 band. Similar results were seen in the 48 fibre samples examined. Similarly, full autolytic activation of CAPN3 in human muscle samples by exposure to 500 µM Ca^2+^ for 60 min did not result in any noticeable proteolysis of RyR1 (Figure 5B,D), with there being no increase in any RyR1 proteolytic products in the range from 180 to 550 kDa and again no apparent loss of the RyR1 band (*n* = 4, *p* = 0.87, Wilcoxon ranked sum test; Figure 5D). As expected from previous findings [17], the Ca^2+^ treatment did cause close to complete (>95%) proteolysis of junctophilin-1 (JP1) from its full-length ~90kDa form to a ~75 kDa product, the latter being largely (>75%) diffusible and moving to the cytosolic/Cyt fraction; virtually identical results were seen with the samples from all four individuals examined (*p* < 0.05, paired *t*-test and two-way ANOVA; Figure 5F,G). Very similar results were also found in Ca^2+^-treated rat skinned fibres using the same JP1 antibody (not shown). The Ca^2+^ treatment caused autolytic activation of CAPN1 (shift from 80 kDa full length to 78 and 76 kDa autolytic products, Figure 5C) as well as proteolysis of some titin at the Z-disk region, as detected by a ~3-fold increase in low molecular weight (~30–40 kDa) diffusible fragments of titin (*n* = 4, *p* = 0.02, paired *t*-test; Figure 5E).

## 3. Discussion

### 3.1. CAPN3 Is Not a Na^+^-Dependent Protease in Physiological Circumstances

It had been previously shown that in certain in-vitro conditions, exposure to a high [Na^+^] caused autolysis of CAPN3, leading to it being said that CAPN3 is an intracellular Na^+^-dependent protease [9]. However, the Na^+^-dependent autolysis of CAPN3 in that previous study was seen in very non-physiological ionic conditions, in particular with high [Na^+^] and no K^+^. The findings of the present study confirm that CAPN3 does undergo autolysis in the presence of a high [Na^+^] (162 mM Na^+^ with zero K^+^) (Figure 1, Figure 2, Figure 4 and Figure 5), but such autolysis did not occur in the presence of even 36 mM Na^+^ (Figure 1, Figure 2, Figure 3, Figure 4 and Figure 5), which is greater than the highest level that the [Na^+^] would reach with extreme physical activity or direct stimulation (25–30 mM [10,11]), provided that K^+^ was present as the predominant intracellular cation, as is the case in all physiological circumstances in muscle. Further, it was found that CAPN3 also autolysed if there was no K^+^ present and the primary cation was Tris, with only 16 mM Na^+^ present (Figure 2), which is consistent with the autolysis resulting primarily from the absence of all K^+^ rather than being caused by the presence of a high [Na^+^] per se. The same earlier study [9] also reported that in cultured muscle cells, application of either monensin, an ionophore specific for monovalent metal ions (Na^+^ >> K^+^), or ouabain, a specific inhibitor of the Na/K-ATPase, caused increased autolysis of CAPN3. Both of these treatments would have been expected to have raised the intracellular [Na^+^] in the cultured muscle cells. However, that study also noted that the intracellular [Ca^2+^] in the cells also increased with these treatments [9], as would be expected to occur with the depolarizing action of the ensuing changes in the [Na^+^] and [K^+^] gradients and membrane permeability; consequently, the CAPN3 autolysis that was observed is well explained by the known autolytic effects of small rises in intracellular [Ca^2+^] [6].

The fact that CAPN3 autolyses in the presence of Na^+^ when all the normal K^+^ is absent might be due in part to the CAPN3 molecule adopting an aberrant confirmation in such non-physiological conditions. However, the primary explanation is probably that the Na^+^ ions are able to bind and act at a site at which normally only Ca^2+^ acts, which would be consistent with the findings of the CAPN3 mutation experiments in that earlier study [9]. Such altered binding at sites that are normally Ca^2+^-specific has been previously documented for the ryanodine receptor (RyR1), where competitive binding of Na^+^ at a normally specific Ca^2+^ is augmented in the absence of K^+^ [18].

### 3.2. Activation of CAPN3 in Physiological Circumstances

Thus, it appears that under the normal physiological conditions within muscle cells, CAPN3 is not a Na^+^-dependent protease, but instead simply a Ca^2+^-dependent protease, just like the ubiquitous calpains, CAPN1 and CAPN2. In such conditions, CAPN3 has a high Ca^2+^-sensitivity, with half-maximal autolysis in human muscle homogenates occurring at ~1 µM free Ca^2+^ over a 60 min exposure at room temperature (Figure 3B). The present study also provided the first direct comparison with the Ca^2+^-dependence of CAPN1 autolysis occurring simultaneously in the same tissue, which was shown to be ~5 times less sensitive to the free [Ca^2+^] (Figure 3B). During normal muscle activity, Ca^2+^ release from the SR gives rise to comparatively large Ca^2+^ transients, with the intracellular [Ca^2+^] rising from its normal resting level of ~50 nM to transiently reach as high as ~2–20 µM in the cytoplasm as a whole [19,20] and likely higher closer to the Ca^2+^ release channels. However, these Ca^2+^ transients, even the during repeated tetani of normal exercise, are evidently too brief to result in measurable autolysis and activation of either CAPN1 and CAPN3 in trained human subjects performing prolonged cycling [13]. Of note, the only physiological circumstance that has been shown to result in endogenous CAPN3 activation in skeletal muscle is 3–24 h following eccentric exercise (i.e., lengthening contractions) in humans [21,22] and rodents [23,24]. Such exercise results in a small but very prolonged increase in the resting [Ca^2+^] in the muscle cells (to >200 nM for 24–48 h) [25]. These findings appear in good accord with the observation here (Figure 3A) and the previous finding [6] that a sustained increase of intracellular [Ca^2+^] to 200 nM for 60 min caused autolysis of ~20% of the CAPN3.

### 3.3. Diffusibility of Autolysed CAPN3

It was found here in both rat and human muscle tissue that autolysis of CAPN3, by either the high [Na^+^]–zero K^+^ treatment or exposure to 500 µM Ca^2+^, resulted in CAPN3 becoming diffusible, that is, unbinding from titin and partitioning into the cytoplasm (Figure 2, Figure 3, Figure 4 and Figure 5). Interestingly, this contrasts markedly with our previous observations in rat skinned muscle fibres where raising the [Ca^2+^] from 50 nM to 200 nM for 60 min resulted in ~20% autolysis of the CAPN3, with this autolysed CAPN3 seen to remain entirely bound in the fibre (Figures 4A and 5A in [6]). Significantly, in the latter case with the 200 nM Ca^2+^ treatment, the CAPN3 was autolysed only partially to the 60 and 58 kDa forms, whereas in all the cases examined in the present study with Na^+^ or very high [Ca^2+^] treatment the CAPN3 was almost entirely autolysed fully to the 55 kDa form (Figure 2, Figure 3, Figure 4 and Figure 5). This marked difference in diffusibility of the autolysed CAPN3 in the different experiments strongly suggests that CAPN3 only becomes readily diffusible from titin once the autolysis has produced complete excision of the IS1 region in CAPN3. Such unbinding and ready diffusibility of the fully autolytically activated CAPN3 would greatly increase its range of potential proteolytic targets.

### 3.4. No Evidence of RyR1 Proteolysis upon CAPN3 Activation

It has been reported that the skeletal muscle ryanodine receptor (RyR1) is proteolyzed by an endogenous muscle protease claimed to be CAPN3 [12], although this was only shown in non-physiological ionic conditions. The great majority of CAPN3 in skeletal muscle is associated with the contractile apparatus, primarily bound at the N2A line on titin [6,7,8]. It has been reported that CAPN3 is associated with the RyR1 at the triad junction [26]. This could only be a very small percentage of the total CAPN3 in muscle, because treatment of skinned muscle fibres with Triton X-100 for 10 min washes out ~90% of the RyR1s and other SR proteins with little concomitant loss of CAPN3 [6]. Nevertheless, given that the triad junction is positioned close to the N2A line on titin, it is entirely plausible that CAPN3 could proteolyze RyR1, particularly given that the autolysed form of CAPN3 was seen in the present study to be diffusible and readily move to the cytoplasm (Figure 2, Figure 3, Figure 4 and Figure 5). However, it was found here that autolytic activation of CAPN3, by either high [Na^+^] with zero [K^+^] or 500 µM Ca^2+^, did not cause any detectable proteolysis of RyR1 in muscle fibres in physiological ionic conditions (Figure 5A,B). Thus, it is possible that the reported proteolysis of RyR1 by CAPN3 [12] was the result of the non-physiological conditions of the assay. Alternatively, it seems quite possible that the endogenous protease causing the RyR1 proteolysis in those experiments was actually CAPN1, not CAPN3, because exogenous CAPN1 has been previously shown to proteolyze RyR1 [27]; the study by Shevchenko et al. [12] reported that the proteolytic activity of the endogenous protease was blocked by both calpain inhibitor 1 (N-acetyl-Leu-Leu-norleucinal) and leupeptin, and these inhibitors block proteolytic activity by CAPN1 and CAPN2 but not CAPN3 [28,29]. The primary reason that the authors attributed the proteolytic activity to CAPN3 was because of the apparent high Ca^2+^-sensitivity of the action [12]. However, this apparent high Ca^2+^-sensitivity might well have been a result of the non-physiological ionic conditions, particularly given that the authors also reported that the protease was half maximally activated by 22 µM free Mg^2+^ [30], which clearly indicates aberrant activation, as there is normally ~1 mM free Mg^2+^ intracellularly present in muscle cells.

Although the 500 µM Ca^2+^ treatment of human muscle homogenates did not cause any detectable proteolysis of the RyR1, it did cause almost complete proteolysis of JP1 from its native full-length ~90 kDa form to a ~75 kDa proteolytic product (Figure 5B), in accord with the original description of JP1 proteolysis [17]. The findings here using the recategorized JP1 antibody are virtually identical to the original observations found using a JP1 antibody (40–5200 to mid-region of JP1, [17]), with both antibodies showing that a loss of the 90 kDa band was accompanied by a quantitatively matching appearance of the 75 kDa band in human and rat muscles, which is consistent with: (1) the antibody detecting JP1 and (2) proteolysis of full-length JP1 being accompanied by an equimolar formation of the 75 kDa product. Furthermore, both studies found that the 75 kDa product was readily diffusible in the cytoplasm (Figure 5B here and Figure 3E in [17]). These findings, which correspond well with matching data in rat and mouse muscle fibres showing that the formation of the 75 kDa diffusible product was accompanied by the appearance of a non-diffusible 15 kDa C-terminal product of the JP1 [17], indicate that the proteolysis of JP1 is the result of a single cleavage close to the C-terminal of JP1, leaving the latter embedded in the SR and the ~75 kDa N-terminal product diffusible in the cytoplasm. These results contrast markedly with a recent report that Ca^2+^-induced proteolysis of JP1 in human muscle produces a ~44 kDa C-terminal product that can translocate to the nucleus [31]. The proteolysis of JP1 seen here with high [Ca^2+^] treatment (Figure 5B) was most likely due to a proteolytic action of CAPN1, as the treatment here caused autolysis of CAPN1 (Figure 5C) and CAPN1 has been previously shown to undergo autolytic activation in a concentration-dependent manner in tandem with the JP1 proteolysis [17]. Furthermore, complete knockout of CAPN3 was found to have no noticeable effect on the Ca^2+^-induced proteolytic disruption of excitation–contraction coupling in mouse skeletal muscle fibres [32].

Although CAPN1 has previously been found to be able proteolyze RyR1 [27], this was when exogenous CAPN1 was added to an isolated SR vesicle preparation. Here, it was found that when the endogenously present CAPN1 was activated by Ca^2+^, it did not cause such RyR1 proteolysis, possibly because of both the relative amount and accessibility of the CAPN1 and also the fact that the endogenous CAPN1 inhibitor calpastatin would have been present in the whole muscle homogenates used here.

In summary, this study has shown the importance of investigating physiological processes under physiological conditions. It had been previously concluded that CAPN3 was activated not only by Ca^2+^ ions but also by Na^+^ ions. However, those experiments were conducted under non-physiological conditions in which all the K^+^ ions normally present in a muscle cell were completely absent. Here, it was confirmed with both rat and human muscle that CAPN3 is activated by exposure to a high [Na^+^] in the absence of K^+^, but most importantly it was further shown that this did not happen even at high physiological [Na^+^] when there was also a normal intracellular [K^+^] present. Hence, it is apparent that CAPN3 is like the ubiquitous calpains, CAPN1 and CAPN2, in that it is activated only by increases in intracellular [Ca^2+^]. It was further shown that, when examined in the same conditions, CAPN3 was ~5 times more sensitive to Ca^2+^ than CAPN1, with 50% of CAPN3 undergoing autolytic activation in 60 min at ~1 µM Ca^2+^, whereas ~5 µM Ca^2+^ was required for a similar activation of CAPN1. Additionally, it was found that after it was autolysed, CAPN3 dissociated from its tight binding on titin and became readily diffusible, but only if the autolysis led to complete removal of the IS1 inhibitory peptide within CAPN3. Such liberation from its binding site on titin would potentially enable CAPN3 to access and proteolyze far more target proteins than when it was bound on titin. The relevant physiological and pathological targets of CAPN3, however, remain unknown. Contrary to a previous report, activation of CAPN3, either by raised [Ca^2+^] or Na^+^ treatment, did not cause proteolysis of the skeletal muscle Ca^2+^ release channel (ryanodine receptor), RyR1, in physiological ionic conditions. In contrast, treatment of human muscle homogenates with high [Ca^2+^] caused autolytic activation of CAPN1, which was accompanied by, and likely caused, proteolysis of some titin and complete proteolysis of junctophilin (JP1, full length ~95 kDa), the latter generating an equimolar amount of a diffusible ~75 kDa N-terminal JP1 fragment. Significantly, this near complete proteolysis of JP1 was not accompanied by noticeable proteolysis of RyR1 despite the close physical association of these proteins at the triad junction in muscle, showing the specificity of action of CAPN1.

## 4. Materials and Methods

### 4.1. Chemicals

All chemicals were purchased from Sigma-Aldrich (Sydney, Australia) unless stated otherwise.

### 4.2. Solutions

All solutions used in these experiments had physiological levels of ionic strength and pH. In order to investigate the effects of [Na^+^] on CAPN3 activity and localisation, three different solutions with various concentrations of [Na^+^] and [K^+^] were used. The first two solutions were physiological-like, K^+^-based solutions, with the first having a [Na^+^] of 36 mM, which is slightly above the maximum level reached with extreme muscle activity (‘K^+^-based solution with 36 mM Na^+^’), and the second with [Na^+^] at 10 mM, close to that in a rested muscle fibre (‘K^+^-based solution with 10 mM Na^+^’). Specifically, the first of these solutions, which was the same as that used in many previous studies on skinned muscle fibre function from this laboratory [33], contained (in mM): 126 K^+^, 36 Na^+^, 1 free Mg^2+^ (10.3 total Mg) 90 HEPES, 50 EGTA, 8 ATP and 10 creatine phosphate (CrP), at pH 7.10 and 295 ± 10 mosmol/kg H_2_O. This solution was strongly buffered with EGTA to keep the free [Ca^2+^] very low (<10 nM) at all times. The second physiological solution was very similar but contained 137 mM K^+^, 10 mM Na^+^ and 15 mM Tris (the Tris was present because Tris_2_CrP was used to replace most Na_2_CrP, as the K^+^ salt of CrP is unstable). The third solution (‘Na^+^-based solution with no K^+^’) was a non-physiological solution exactly matching the K^+^-based solution with 36 mM Na^+^ but with Na^+^ replacing all the K^+^ (i.e., 162 mM Na^+^ and zero K^+^) (see [33]). In addition, a Tris-Cl solution with 16 mM Na^+^ was made by mixing a 150 mM Tris-Cl solution (pH 6.8) in a ratio of 9:1 (vol:vol) with the Na^+^-based solution with no K^+^; this Tris-Cl solution had a lower final [ATP] (0.8 mM) and [CrP] (1.6 mM) than the above solutions, but the free [Ca^2+^] was still strongly buffered to very low levels with 5 mM EGTA.

For experiments investigating the Ca^2+^-dependence of CAPN3 autolysis, a further K^+^-based solution was made with the free [Ca^2+^] strongly buffered at a relatively high level (20 µM). This solution was virtually identical to the K^+^-based solution with 36 mM Na^+^, but contained 49.5 mM Ca-EGTA and 0.5 mM free EGTA to give a free [Ca^2+^] of ~20 µM (pCa = −log_10_ [Ca^2+^] ~4.7) and had 8.12 mM total magnesium to maintain the free Mg^2+^ at 1 mM (see [33]). These two matching solutions were mixed in a ratio of 1:1, 1:3, 1:10 to produce additional solutions with the [Ca^2+^] strongly buffered at 0.2 µM, 0.5 µM and 2 µM, respectively. In addition, a solution with a free [Ca^2+^] of ~500 µM was prepared by adding 40 μL of 250 mM CaCl_2_ solution to 5 mL of the high [Ca^2+^] solution.

The denaturing solution was 3 × SDS loading buffer, containing 0.125 M Tris-Cl, pH 6.8, 4% SDS, 10% glycerol, 4 M urea, 10% mercaptoethanol and 0.001% bromophenol blue.

### 4.3. Collection of Samples

Both human and rat muscle samples were used in this study.

Male Sprague-Dawley (4–6 months old, *n* = 9) and male Long Evans (6–9 months old, *n* = 2) rats were euthanized by overdose with inspired isoflurane (4% vol/vol), as approved by the La Trobe University Animal Ethics Committee (AEC14/33, approved 18/8/2014 and AEC06/09, approved 1 April 2006). Extensor digitorum longus (EDL) muscles were dissected for immediate freezing or for single fibre collection.

The human skeletal muscle samples were obtained from the *vastus lateralis* muscle from *n* = 11, individuals aged 14–43 years old under sterile conditions using local anaesthesia (Xylocaine) and a Bergstrom needle modified for manual suction [34,35]. Samples were a subset of two larger studies approved by La Trobe University Human Ethics Committee (FHEC09/R43, approved 18 June 2009 and UAHPEC 2010/314, approved 23 March 2013) and conducted in accordance with the Declaration of Helsinki.

### 4.4. Rat Samples and Single Fibre Diffusion Experiments

CAPN3 autolysis was examined in rat EDL muscle following exposure to the K^+^-based solution with 10 mM Na^+^, the K^+^-based solution with 36 mM Na^+^, or the Na^+^-based solution with no K^+^, for various lengths of time at room temperature (RT). Frozen EDL muscles from 2 rats were cryosectioned (10 µm) and ~20 sections placed in one of the three solutions (70 μL) for the desired time (1, 15, 30, 60 or 90 min) and then mixed (2:1 vol:vol) with 3 × SDS loading buffer. These denatured muscle samples were stored for later analysis for CAPN3 autolysis by Western blotting.

For single fibre collection, excised EDL muscles were blotted dry on filter paper, pinned at resting length under paraffin oil and kept at ~10 °C. Single EDL fibres were mechanically skinned (*n* = 192 fibres), as previously described [33,36]. To examine CAPN3 diffusibility, groups of skinned fibre segments (3 fibres per tube) were incubated in 10 µL of treatment solution (Na^+^-based solution with no K^+^, K^+^-based solution with 36 mM Na^+^ or Tris-Cl solution with 16 mM Na^+^ and no K^+^) for 60 min at RT. The skinned fibre portions (F) were collected into a fresh tube with 10 µL of the given treatment solution mixed with 5 µL of 3 × SDS loading buffer. The treatment solution (W) was collected separately and mixed with 5 µL of 3 × SDS loading buffer. The fibre (F) and matching wash (W) solutions were stored at −20 °C until Western blot analysis.

### 4.5. Human Samples, Calcium-Treated Muscle Homogenates and Crude Fractionation Experiments

CAPN3 autolysis was examined in frozen human muscle samples in a similar way to that described above for rat tissue. Cryosections from 6 individuals were cut (~6 × 10 µm) and immediately placed into ~70 µL of ice-cold solution of known [Ca^2+^] (see above). Samples were incubated for 60 min at RT, with vortexing of samples 3–4 times and then mixed (2:1 vol:vol) with 3 × SDS loading buffer. Samples were stored at −20 °C until Western blot analysis.

Unfractionated muscle homogenate samples were prepared by homogenising 5–10 mg of frozen human skeletal muscle tissue (*n* = 4) in either the K^+^-based solution with 36 mM Na^+^ at <10 nM Ca^2+^ (Control) or the matching solution with 500 µM free Ca^2+^ (Ca^2+^-treated; 1:30 wt:vol) using a handheld polytron homogeniser as described previously [14]. Control and Ca^2+^-treated samples were incubated at RT for 60 min, then a portion (~100 µL) was removed (whole homogenate, Wh). The remaining portion (~100 µL) was centrifuged at 1000× *g* for 10 min at 4 °C. The supernatant (100 µL), representing a crude cytosolic fraction (Cyt), was removed, and the pellet (Pel), representing the remaining bound proteins (membrane and myofibrillar/cytoskeletal), was resuspended in 100 µL of the same originally used solution, i.e., Control or Ca^2+^-treated. Following fractionation, each fraction was mixed (2:1 vol:vol) with 3 × SDS loading buffer, vortexed 3–4 times (~10 s) and incubated for 60 min at RT. Samples were stored at −80 °C until Western blot analysis.

To determine the effects of the high Na^+^/zero K^+^ treatment on CAPN3 localisation, samples were prepared as described above except using either the control K^+^-based solution with 36 mM Na^+^ or the Na^+^-based with no K^+^; when the Cyt fraction was removed after ~20 min, the Pel was treated with the 100 µL of the same solution containing 1% Triton-X 100 for 10 min and then centrifuged at 14,000× *g* for 10 min at 4 °C to obtain the membrane (Me) fraction, and the final pellet (Csk fraction) containing the nuclear and myofibrillar/cytoskeletal proteins was resuspended in 100 µL of the original solution (see [14]). Note that the aforementioned study [14] showed that the nuclei were not separated by the Triton-X 100 treatment and remained in the cytoskeletal/Csk fraction. Note also that in the above fractionation experiments, nothing was discarded and all the fractions were diluted to the same final volume, so that when run side by side on a Western blot the band densities in the Cyt, Me and Csk fractions indicate the quantitative distribution between the different fractions, and for any given band the sum of the band densities across the three fractions would be expected to equal that in the Wh sample (at least for the case where the calibration procedure described below indicated that the band density was approximately directly proportional to the amount of protein present).

### 4.6. Western Blotting

Western blotting was performed as previously described, with calibration curves run on every gel used to ascertain the relationship between density and amount of protein [14,37,38]. Proteins were separated on 4–15% gradient or 10% Criterion TGX Stain Free SDS-PAGE gels (Bio-Rad Laboratories, Hercules, CA, USA), which can be imaged following electrophoresis and the relative amount of tissue loaded in each lane can be determined from the UV activated image of the Stain Free gel, using a ChemiDoc MP Imager (Bio-Rad) and Image Lab software (v5.2.1, Bio-Rad). In addition, 4–12% Criterion XT Bis-Tris gels (Bio-Rad) were used to examine RyR1 protein amounts in rat EDL fibres and human whole muscle samples. Various controls were used, as described in figure legends. The primary antibodies used were: mouse anti-CAPN1 (C0355, Sigma-Aldrich), rabbit anti-CAPN2 (C3989, Sigma-Aldrich), mouse anti-CAPN3 (12A2, Novocastra), rabbit anti-SERCA2a (A010-20, Badrilla), mouse anti-GAPDH (Ab8245, Abcam), mouse anti-RyR1 (34C, Developmental Studies Hybridoma Bank), mouse anti-JSRP1 (NB120-2875 clone VF1c, Novus Biologicals) and mouse anti-titin N-terminal fragments (H00007273-M06, Abnova), which is designed to target the Z-disk region of titin [39]. It is important to note that the JSRP1 protein is a 45 kDa protein and this commercially available antibody was made with antigen-targeting rabbit SR triads, but the antibody does not detect 45 kDa JSRP1. Instead, it detects a 90 kDa protein which is evidently JP1, identified as the protein sensitive to Ca^2+^-dependent proteolysis that we have previously shown for JP1 [24]. From here we will refer to this protein and antibody as anti-JP1 (see Section 2). Following imaging of CAPN3, the membrane was stripped for 15 min at 37 °C using Restore Western Blot Stripping Buffer (Thermo Fisher Scientific, Rockford, IL, USA), washed in Tris-buffered saline with Tween-20 (TBST) for 10 min and then incubated in mouse anti-JP1 antibody (see Figure 5B).

### 4.7. Statistical Analyses

Individual and mean (+standard deviation) data are presented unless specified differently in the figure legend. Statistical analyses were performed using GraphPad Prism 9.1.0 (San Diego, CA, USA). Parametric tests were performed using the Shapiro–Wilk test (*p* < 0.05) except when data did not pass the test for normal distribution, in which case a non-parametric test was used. The Ca^2+^-dependence of both CAPN1 and CAPN3 was determined by plotting the log_10_ of [Ca^2+^] (pCa) (*x*-axis) against the autolysis level of CAPN (*y*-axis) and fitted with sigmoidal dose–response curves. The Akaike information criterion (AIC) method was used to compare the two curve models for CAPN autolysis data, based on the goodness-of-fit and the number of parameters in the model. A Student’s paired *t*-test (two-sided) was used to compare CAPN3 autolysis levels in whole human muscle samples prepared using either K^+^-based solution with 36 mM Na^+^ or the Na^+^-based solution with no K^+^. Paired comparison tests were also used to examine the effect of [Ca^2+^] (<10 nM Ca^2+^ vs. 500 µM Ca^2+^) on the amount of RyR1 (Wilcoxon ranked sum test test) and on titin N-terminal fragments (paired *t*-test). A repeated measures or mixed effects model, two-way ANOVA along with the appropriate multiple comparison test was used to examine the effect of the solution base (i.e., K^+^-based solution with 36 mM Na^+^ or Na^+^-based solution with no K^+^) and [Ca^2+^] (<10 nM Ca^2+^ vs. 500 µM Ca^2+^) on CAPN3 autolysis and CAPN3, CAPN1 and JP1 distribution in human fractionated samples from whole muscle homogenates. Statistical significance was set at *p* < 0.05.

## 5. Conclusions

As detailed fully throughout the manuscript, this study has shown the importance of investigating physiological processes under physiological conditions. 

## Figures and Tables

**Figure 1 ijms-24-09405-f001:**
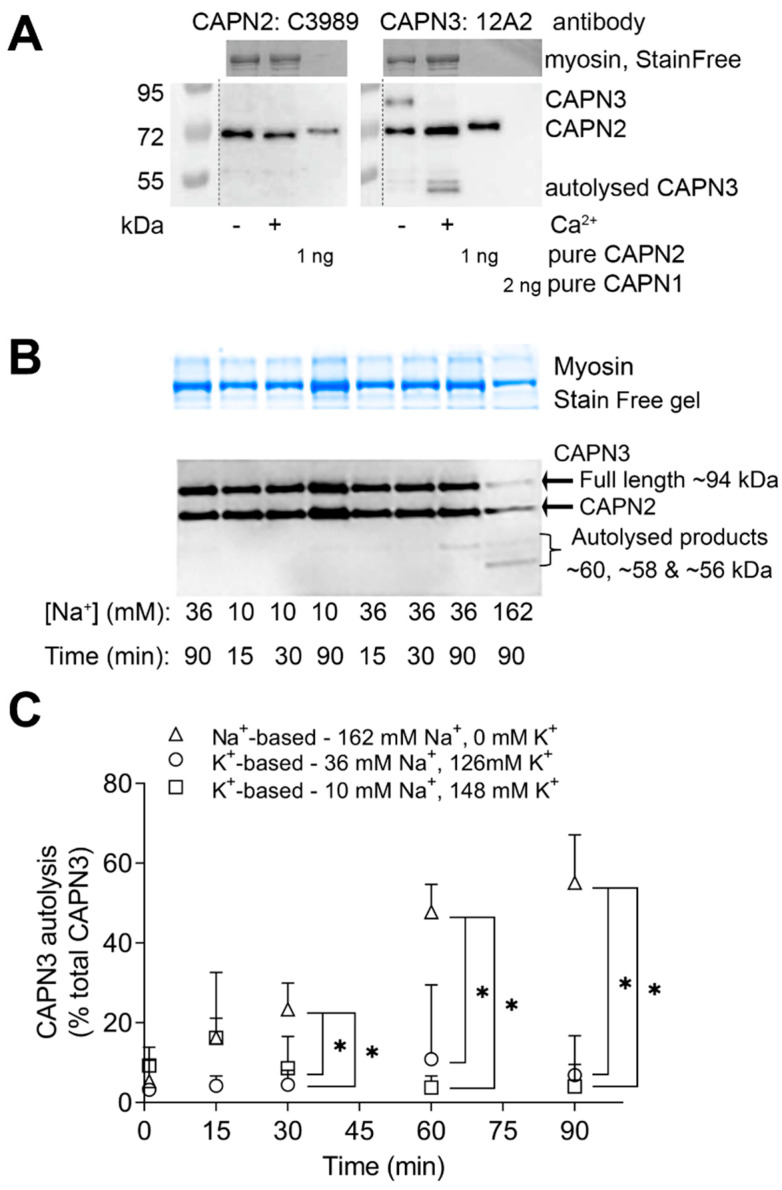
Non-physiological high [Na^+^] with zero [K^+^] results in CAPN3 autolysis and activation. (**A**): Long Evans rat whole muscle homogenates were treated with or without 5 mM Ca^2+^ for 10 min and run on Western blots, with total muscle protein indicated by myosin on the Stain Free gel. Samples probed using anti-CAPN3 antibody (12A2) (right hand panel) detected full-length CAPN3 in the untreated samples and autolysed CAPN3 bands at 60, 58 and 55 kDa following Ca^2+^ treatment. The 12A2 antibody also detected a band at ~82 kDa. This band was evidently CAPN2, as the 12A2 antibody was found to detect exogenous CAPN2, and a specific anti-CAPN2 antibody (C3989) identified the same 82 kDa band in the muscle samples (left hand panel). The 12A2 antibody did not detect exogenous CAPN1. (**B**,**C**): Full-length and autolysed CAPN3 in rat muscle samples treated for 0–90 min with solutions with various concentrations of Na^+^ and K^+^ but were otherwise matching (see Section 4). Non-physiological (Na^+^-based) solution 162 mM Na^+^ with zero K^+^ (triangles); solution with [Na^+^] at normal intracellular level of resting muscle (10 mM) and 137 mM K^+^ (squares); solution mimicking intracellular [Na^+^] in a highly exercised muscle (36 mM Na^+^, 126 mM K^+^) (circles). (**B**): Total protein in samples separated on 10% Criterion TGX Stain Free gels and the total protein observed on the UV-activated Stain Free gel (top, myosin) followed by Western blotting for CAPN3 (bottom). (**C**): the amount of autolysed CAPN3 expressed as a percentage of the total CAPN3 present in the given sample (i.e., sum of 60, 58 and 55 kDa bands divided by sum of 94, 60, 58 and 55 kDa bands, as %), for the different treatments as detailed above. *n* = 3–6 homogenates per time point. * Different from the Na^+^-based solution (*p* < 0.05, two-way ANOVA). CAPN3 autolysis levels did not differ between the two physiological K^+^-based solutions at any time point (*p* > 0.05, two-way ANOVA).

**Figure 2 ijms-24-09405-f002:**
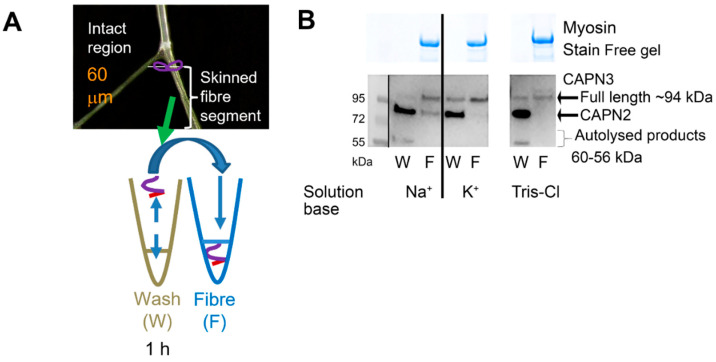
CAPN3 in mechanically skinned rat EDL fibres autolyses in solutions lacking all K^+^. (**A**): Mechanically skinned Sprague-Dawley rat EDL fibres were isolated (3 fibre segments per sample) and placed into either the Na^+^-based solution with no K^+^, the K^+^-based solution with 36 mM Na^+^ or the Tris-based solution with 16 mM Na^+^ and no K^+^ for 60 min and then collected in a separate tube. Each fibre sample (F) and its matching wash (W) solution were subsequently analysed by Western blotting. (**B**): Western blots showing full-length (~94 kDa) and autolysed (~60–56 kDa) CAPN3 present in W/F sets described in (**A**). Myosin on Stain Free gel visually indicates the total amount of protein in the given fibre sample. A solid line separates the two solution bases. A space indicates noncontiguous lanes. Experiments repeated several times (*n* = 192 fibres used) from 9 rats produced similar results.

**Figure 3 ijms-24-09405-f003:**
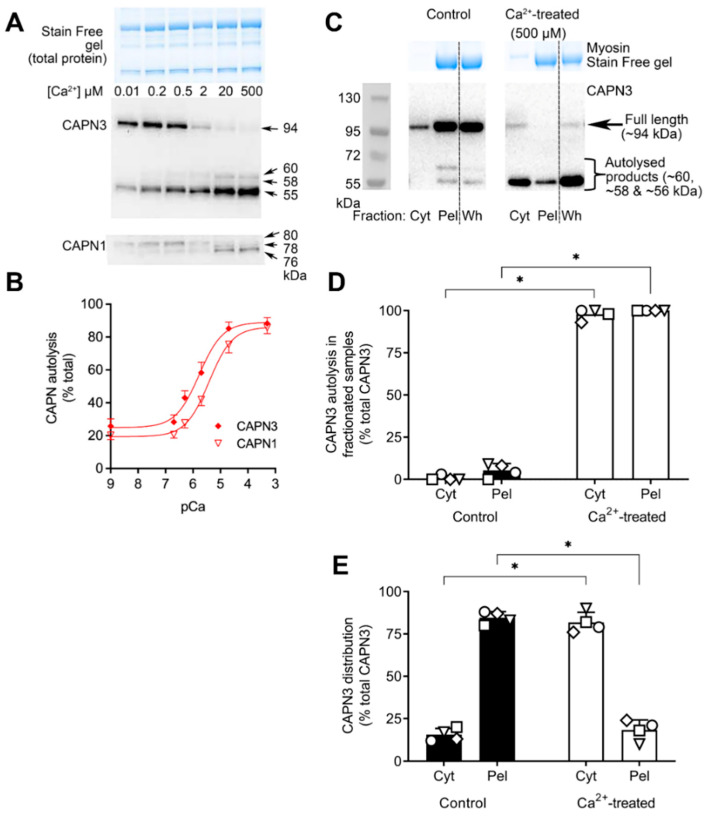
Calcium-dependence of CAPN3 and CAPN1 autolysis, and redistribution of autolysed CAPN3 to cytosol. Human muscle samples (*n* = 6) cryosectioned and immediately treated with the standard K^+^-based solution (126 mM K^+^, 36 mM Na^+^) containing 0.01–500 µM free Ca^2+^ at RT for 60 min. (**A**): Total protein in samples separated on 10% Criterion TGX Stain Free gels and observed on the UV-activated Stain Free gel (top), followed by Western blotting for CAPN3 (middle) and then CAPN1 (bottom). Molecular weights of proteins shown on the right hand side, with the [Ca^2+^] in µM shown for each lane. (**B**): Amount of autolysed CAPN3 expressed as a percentage of total CAPN3 present in given sample (i.e., sum of 60, 58 and 55 kDa bands divided by sum of 94, 60, 58 and 55 kDa bands, as %) and that of autolysed CAPN1 expressed as a percentage of the total CAPN1 present in given sample (i.e., sum of 78 and 76 kDa bands divided by sum of 80, 78 and 76 kDa bands, as %). The Ca^2+^-dependence of both CAPN1 and CAPN3 was determined by plotting the log_10_ of the [Ca^2+^] (pCa) (*x*-axis) against the autolysis level of CAPN (*y*-axis) and fitted with a sigmoidal dose–response curve. The Akaike information criterion (AIC) method was used to compare the two curve models for CAPN autolysis data, based on the goodness-of-fit and number of parameters in the model. Analysis indicated there was a 99.5% probability that the data support different curves rather than one curve for both data sets. Mean + SEM for data from *n* = 6 individuals. (**C**–**E**): Human muscle samples homogenised in the K^+^-based solution with 36 mM Na^+^ either with [Ca^2+^] buffered at <10 nM or raised to 500 µM and kept at RT for 60 min. Following removal of a portion to represent whole, unfractionated muscle (Wh), samples were spun (1000× *g*, 10 min) to obtain crude cytosolic fraction (Cyt) and pellet (Pel) fractions (see Section 4). (**C**): Representative Western blots showing CAPN3 in Cyt and pellet fractions, alongside the unfractionated whole/Wh muscle sample. Myosin on the Stain Free gel visually indicates total amount of protein in the given sample. A dashed line separates the Cyt and Pel samples from the corresponding Wh sample. A space indicates noncontiguous lanes between the marker and the Cyt lane. A space is also provided to visually separate Control and Ca^2+^-treated samples, which were run contiguously. (**D**): Autolyzed CAPN3 in Cyt and Pel fractions, expressed as a percentage (+SD) of the total CAPN3 in that fraction (i.e., sum of 60, 58 and 55 kDa bands in Cyt divided by sum of 94, 60, 58 and 55 kDa bands in Cyt, as %). (**E**): CAPN3 distribution determined as density of the total bands in the Cyt and Pel samples under Control (black bars) and Ca^2+^-treated (white bars) conditions, expressed as a percentage of total in the given fraction (i.e., Cyt + Pel = 100%). * *p* < 0.05, different from Control, *n* = 4 individuals, two-way ANOVA.

**Figure 4 ijms-24-09405-f004:**
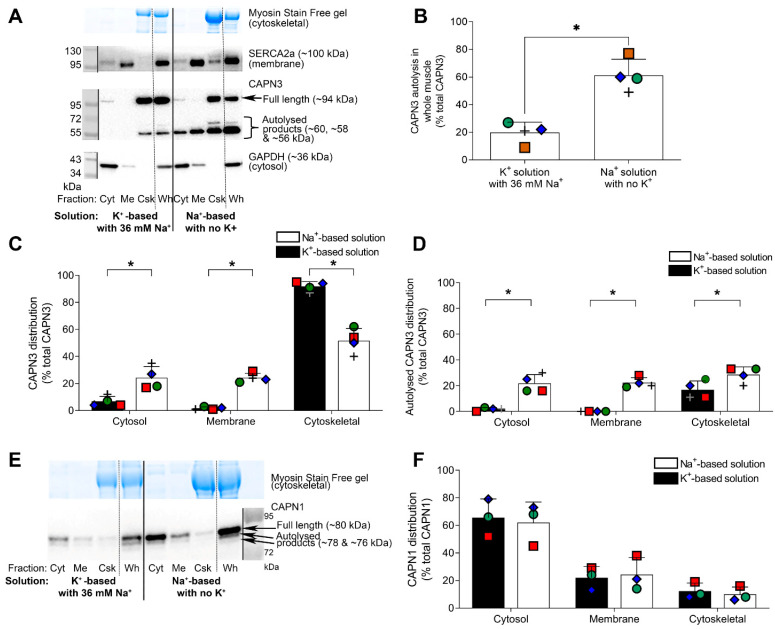
Na^+^ treatment of human skeletal muscle samples causes autolysis and redistribution of CAPN3 but not CAPN1. (**A**): Representative Western blot showing CAPN3 autolysis and localization in muscle samples prepared in physiological K^+^-based solution with 36 mM Na^+^ or Na^+^-based solution with no K^+^ for one human muscle sample (shown as red squares in all graphs). The lanes marked Wh shows the ‘whole’ (unfractionated) samples, a portion of which was separated into three fractions: a cytosolic/Cyt fraction (GADPH present), a membrane/Me fraction (SERCA2a present) and the remaining cytoskeletal/Csk fraction (myosin seen on the Stain Free gel). Vertical dotted line on Western blot visually separates the lanes with the fractions from the matching unfractionated (Wh) muscle sample. Solid line separates the two solution treatments. Molecular weight markers shown on the left, with the observed band size (kDa) shown on the right. (**B**): mean data ofCAPN3 autolysis in the unfractionated muscle samples (Wh) prepared in two solutions; values expressed as mean percentage (+SD) of total CAPN3 in Wh lane [i.e., % autolysed bands = autolysed bands/sum of all CAPN3 bands].Same coloured symbol represents same individual in all panels. (**C**): Distribution of CAPN3 in subcellular fractions prepared from the same whole muscle samples shown in (**B**), expressed as the mean percentage (+SD) of total CAPN3 in all fractions [e.g., % Cyt = all CAPN3 bands in Cyt lane/sum of all CAPN3 bands in Cyt + Me + Csk lanes]. (**D**): Distribution of autolysed CAPN3 in the same subcellular fractions as shown in (**C**), expressed as mean percentage (+SD) of total CAPN3 in all fractions [e.g., % autolysed CAPN3 in Cyt = autolysed CAPN3 bands in Cyt lane/sum of all CAPN3 bands in Cyt + Me + Csk lanes]. * *p* < 0.05 different from Na^+^-based solution (*n* = 4 individuals).Statistical tests used: paired *t*-test (**B**) and two-way ANOVA (**C**,**D** and **F**). Data in (**B**–**D**) attained from mean of two gel runs. (**E**): Western blot showing CAPN1 in same samples as CAPN3 in panel (**A**). (**F**): Mean data showing the distribution of total CAPN1 in the three fractions, examined in three of the same individuals as in panels (**B**–**D**).Such diffusibility would also enable some of the autolysed CAPN3 to translocate to the nucleus. Whilst CAPN3 has been identified in the nuclei of skeletal muscle of healthy individuals [15], that study was not able to distinguish between the full-length and the autolysed forms of CAPN3. In a previous attempt to determine the extent and type of such nuclear translocation, we had explored the biochemical separation of nuclei from the muscle but found that the only way to isolate the nuclear compartment from skeletal muscle required non-physiological solutions that caused aberrant CAPN3 autolysis and mis-localisation [14] and so any physiologically relevant relocation of full-length or autolysed CAPN3 to the nucleus could not be confirmed.

**Figure 5 ijms-24-09405-f005:**
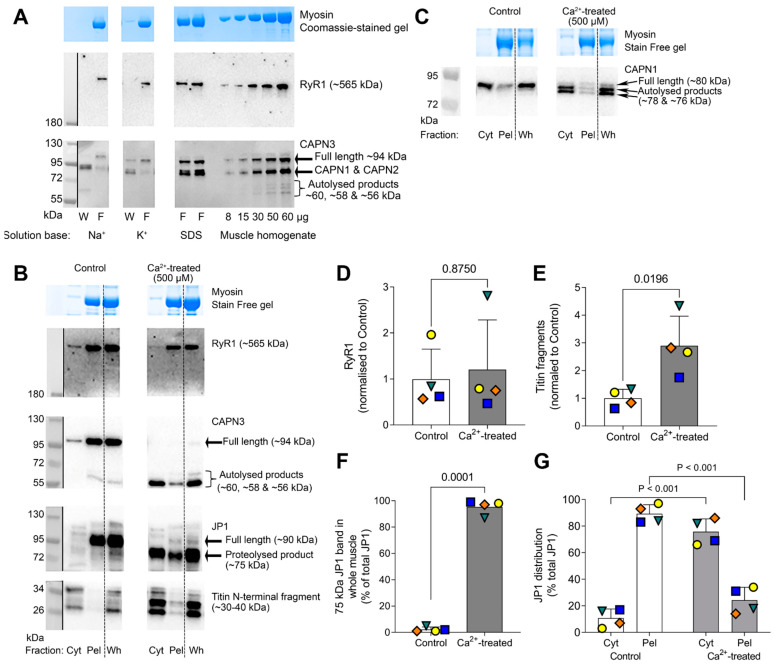
Ryanodine receptor (RyR1) protein is not affected by CAPN3 autolysis following exposure to either non-physiologically high [Na^+^] with zero [K^+^] or 500 µM Ca^2+^. (**A**): Representative Western blots showing RyR1, CAPN1 (first probe) and CAPN3 (second probe) full-length and autolysed forms in present in W/F sets (three Sprague-Dawley rat EDL fibre segments per sample) as described in Figure 2A or fibre segments immediately collected in denaturing solution (SDS). Samples were run alongside a calibration curve of known amounts (i.e., 8–60 µg) of rat EDL homogenate that were prepared using the Na^+^-based solution with no K^+^. Note that CAPN3 blot shows the previous CAPN1 signal and CAPN2 signal because CAPN3 was probed for after first probing for CAPN1, and the CAPN3 antibody also detects CAPN2 in rodent skeletal muscle (see Figure 1A). Myosin seen on the post-transfer Coomassie-stained gel visually indicates the total amount of protein in each sample. A space indicates non-contiguous lanes. These experiments were repeated several times with similar results (*n* = 48 fibres) from 5 rats. (**B**): Representative Western blots of human muscle samples treated with <10 nM (Control) or 500 µM Ca^2+^ for 60 min, showing RyR1, CAPN3, Junctophilin-1 (JP1, see Section 4) and titin N-terminal fragments in the cytosolic (Cyt) and pellet (Pel) fractions, alongside the unfractionated whole/Wh muscle sample (see description in Figure 3 and Section 4). After CAPN3 was imaged, the membrane was stripped for 15 min at 37 °C, then incubated in the mouse anti-JP1 antibody. (**C**): Representative Western blot showing CAPN1 from the same experiments as (**B**). Myosin on the Stain Free gel visually indicates total amount of protein in each sample. A dashed line in B and C separates the Cyt and Pel samples from the matching Wh sample. A space is provided between adjacent Control Wh and Ca-treated Cyt lanes to visually separate Control and Ca^2+^-treated samples. Experiments repeated several times with similar results (*n* = 4 human muscle samples). Mean amount of RyR1 (**D**) and titin fragments (**E**) in whole muscles samples prepared under Control and Ca^2+^-treated conditions, normalized to the mean of the Control samples. Note that each protein density value was expressed relative to the total protein density value in the sample from the UV Stain Free image and then normalized to Control. (**F**): Mean amount of ~75 kDa JP1 band in whole muscles samples expressed as a mean (+SD) percentage of total JP1 detected (i.e., sum of 75 and 90 kDa bands). (**G**): Distribution of JP1 in Cyt and Pel fractions, expressed as mean (+SD) percentage of the total JP1 in all fractions (i.e., Cyt + Pel = 100%) under Control (white bars) and Ca^2+^-treated (grey bars) conditions. Same coloured symbol represents same individual in all panels. *p*-values are provided above each graph (*n* = 4 individuals, paired *t*-test used in (**D**–**F**), and two-way ANOVA used in (**G**)).

**Table 1 ijms-24-09405-t001:** CAPN autolysis in different ionic conditions in skeletal muscle (human and rat).

Ionic Condition:	Ca^2+^ with or without K^+^	Na^+^ without K^+^ Present	Na^+^ with K^+^ Present
CAPN1 autolysis	✓	X	X
CAPN3 autolysis	✓	✓	X

## Data Availability

All data generated or analysed as part of this study are included in this published article or are available from the authors upon reasonable request.

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
