# Peer review of "Calpain-3 Is Not a Sodium Dependent Protease and Simply Requires Calcium for Activation"

_ijms, 2023, doi:10.3390/ijms24119405_

Round 1

Reviewer 1 Report

The presented study was purposed to analyze the impact of ionic conditions (physiological or not) autolytic and diffusion properties of calpain-3 titin-bound protease. The authors also analyzed the catalytical abilities of the molecule regarding the widelu discussed potential CAPN3 targets (RYR1 and JP1). The topic is exciting since CAPN3 has been under study for many years and we are still unaware of the actual functions of the protease. Unfortunately in this papedr the authors again can not answer this question. The paper is well-written and quite clear for the readers.  

The Introduction is well-narrated and quite interesting. No concerns about the methods. I have some concerns regarding the Discussion and Conclusions.

Major

1. It is necessary to improve the structuring of the paper. The issues to some extent are mixed and do not follow any logical order. E.g. Na (non-physiol/physiol/duration) - Ca (the same) - diffusion - substrates. Maybe the paper will gain if the comparative table including species differences is prepared.

2. As for the substrates: What do you think about the actual physiological targets of the CAPN3 during various states (e.g. exercise, disuse, aging etc)? Do you suppose anything?

3. It would be better to compare the ionic-dependence of the CAPN3 and CAPN1 and to discuss them.

4. The author do not give the list of comclusions. This addition will make the paper easier for readers.  

Minor

When discussing fractionation, the author give the same title (Cyt) for cytosolic and cytoskeletal fractions. It is difficult for understanding.

Anyway the manuscript is worth publishing after some corrections and structuring.

Author Response

File attached

Reviewer 2 Report

The manuscript by Wette, Lamb and Murphy reports on the use of gel electrophoresis and western blots to characterize conditions for proteolysis of key proteins in rat and human skeletal muscle fibers. The outcomes are contrary to previous studies that involved less physiologically relevant conditions. Thus, this study is an important addition to the field of skeletal muscle regulation.

Major concerns:

·       Was the Ca2+ dependence of CAPN1 vs CAPN3 autolysis statistically different (Fig 3b)? If so, please provide statistical information. If not statistically significant, please indicate that in the second paragraph for section 3.3, and the legend of figure 3b.

·       Please indicate statistical tests used in the legends of all graphs. This includes data that appear to have no statistical significance, this should be indicated, and the text used should be described in the legend.  

·       Please include the mean data of Figure 5 as graphs, including statistical analysis.

Minor concerns

·        A lot of the text in the methods and Results were difficult to follow. It would be desirable to provide more context in these sections and consider careful wording to enhance reader accessibility. This suggestion should not hold back this manuscript from being accepted for publication, but rather encourages the authors to carefully edit the narrative of the text in the “Materials and Methods” and “Results” sections.

·        Please correct the formatting error on page three. I assume that the text “Rat samples, calcium-treated muscle homogenates and single fibre diffusion experiments” should be a sub heading rather than a dot point. Similarly, the text in the following dot point should not presented as a dot point (page 3).

·       First sentence in section 3.1, please remove word “appear”.

Please thoroughly read over the text before submission. 

Author Response

File attached
